

# Picking the low-hanging fruit:
# Testing new physics at scale with active learning

Juan Rocamonde[1], Louie Corpe[2*], Gustavs Zilgalvis[1],
Maria Avramidou[3] and Jon Butterworth[3]

**1** Department of Pure Mathematics and Mathematical Statistics, University of Cambridge,
Centre for Mathematical Sciences, Wilberforce Rd, Cambridge CB3 0WA, UK
**2** CERN, Esplanade des Particules 1, 1211 Meyrin, Switzerland
**3** Department of Physics & Astronomy, University College London,
Gower St., WC1E 6BT, London, UK

⋆ l.corpe@cern.ch

## Abstract

Since the discovery of the Higgs boson, testing the many possible extensions to the Standard Model has become a key challenge in particle physics. This paper discusses a new method for predicting the compatibility of new physics theories with existing experimental data from particle colliders. Using machine learning, the technique obtained comparable results to previous methods (>90% precision and recall) with only a fraction of their computing resources (<10%). This makes it possible to test models that were impossible to probe before, and allows for large-scale testing of new physics theories.

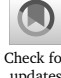

# 1  Introduction

The Standard Model of particle physics is the best description of the building blocks of the universe we have so far. Decades of scrutiny at particle colliders have demonstrated its unprecedented agreement with experimental data. However, it has some major shortcomings: it is unable to account for a variety of observed phenomena, such as neutrino oscillations [1, 2], the presence of dark matter [3] and dark energy [4], or the asymmetry in the quantities of matter and anti-matter [5]. These deficiencies suggest that there must exist physics beyond the Standard Model (BSM). In the last few decades, many theories have emerged in an attempt to account for these deficiencies.

The discovery of the Higgs boson [6–10] completed the search for all the particles predicted by the Standard Model, which had guided experiments so far. Now, for the first time in fifty years, there is no single guiding theory that can motivate new experiments and discoveries. This poses a problem to the field of particle physics: because of the large number of candidate theories to replace the Standard Model, it is unviable to test all of them exhaustively through direct searches at collider experiments. The analyses that need to be designed to do so can take years, and are only suitable for testing a few theories at a time. In addition, they only probe specific values of the parameters of the model, such as the width or mass of their new particles, or the strength of the interaction between them. Extrapolating the results between different regions of parameter space is not always possible.

However, there is a silver lining. The Large Hadron Collider (LHC) [11] data set is one of the largest in scientific history, and is expected to become ten times larger over the next two decades. Hundreds of measurements have already been made in a wide variety of final states. New tools [12] can harness the power of these measurements to efficiently discard those theories that are incompatible with past observations. Even though historic measurements were primarily aimed at understanding Standard Model processes, they implicitly contain information about potential contributions from physics beyond the Standard Model.

Indeed, the addition of new particles and particle interactions in BSM theories often causes deviations from the predictions of the Standard Model in well-measured processes, especially at the energies above the electroweak symmetry-breaking scale currently being probed at the LHC. If the candidate theory were an appropriate model of the process, deviations at these scales would have been detected in past observations, given the large amounts of experimental evidence we have. The $CL_s$ technique [13] can be used to quantify the incompatibility between a theory and

the observed data. By convention, 68% and 95% confidence level (CL) exclusions are considered.

Deviations can also occur at different scales or processes for which we have little or no experimental results; therefore, their corresponding candidate theories cannot be discarded yet given our current pool of measurements. However, as the pools from the LHC and other sources continue to grow, more theories will be testable using this method, reducing the number of candidate theories. This shift in paradigm, from a theory-driven approach to a data-driven approach, will become increasingly important for testing new candidate theories at scale, so that the right models can be targeted in direct searches, and more discoveries can be made.

To achieve this goal, the CONTUR project [12,14] has formalised a methodology for testing a wide array of candidate new-physics models. It simulates model behaviour by generating events using Monte Carlo methods, and compares these results with hundreds of differential cross-section measurements from ATLAS [15], CMS [16] and LHCb [17]. The method applies to arbitrarily complex models, and yields comparable results to direct searches, so long as measurements of relevant final states have been preserved in RIVET routines [18] and made public by the LHC experiments.

While CONTUR obtains results in a few days, which might take years to produce through direct searches, the approach is not computationally viable when attempting to probe the entire parameter space of a given model. The computational complexity scales exponentially with the number of parameters; even a coarse scan with 10 points along each axis becomes unfeasible for models with more than four parameters, and many well-known new-physics models have many more parameters than that. The pMSSM [19], for instance, has 19.

However, not all regions of parameter space are of interest. The goal is to identify the boundaries in the parameter space that separate excluded from non-excluded regions; by convention, the contour lines of 68% and 95% confidence levels, while sampling as few points as possible. It therefore suffices to classify every point in the parameter space into one of 0-68%, 68-95% or 95-100% regions.

This paper presents a method for approaching this classification problem with machine learning, which we have named the CONTUR ORACLE. In this method, a random forest [20] is trained iteratively using partial CONTUR scans to predict the exclusion status of each point in the parameter-space, and new points are iteratively sampled using the uncertainty in these predictions—typically those near the 68% and 95% confidence level contours. This is repeated until the desired levels of accuracy and confidence are achieved.

First, we describe this methodology in detail, providing a motivation for the design choices. Then, we showcase its performance for BSM models studied in past CONTUR publications, obtaining consistent results with a fraction of the computational cost. We also apply the method to a simplified Dark Matter model in five dimensions, with a scan granularity which would have been impossible using only CONTUR. Finally, we discuss avenues for continuation and improvement of this work, and the important role we expect that it will play in the future of particle physics.

## 2 Approach

This section explains the methodology of the ORACLE, including the active learning [21] algorithm, the metrics used to evaluate its performance, the conditions that determine when to stop sampling, and a discussion of the the chosen machine learning method: random forests. The ORACLE is available in the CONTUR library from version 2.2.0.

## 2.1 Motivation

The current CONTUR methodology [12, 14] obtains exclusion results for all the desired points in a new-physics model's parameter space by generating collider events using the Herwig Monte Carlo event generator [22] with a Universal FeynRules Object [23], used to describe the BSM model. Generated signal events are passed through the analysis logic of the preserved RIVET routines [18], and the resulting distributions are compared to the observed results for LHC analyses. The compatibility of a new physics model with the observed data is evaluated using the CL$_s$ technique [13]. Finally, the 95% and 68% CL exclusion contours are drawn. It is worth noting here that the power of the CONTUR method is intimately linked with the availability of RIVET routines for measurements made at the LHC. At time of writing, RIVET routines exist for around 60%, 40% and 7% of relevant ATLAS, CMS and LHCb measurements, respectively. This already represents over 250 measurements in a wide variety of center-of-mass energies and final states, but there are still many key measurements which are not available. We encourage the LHC collaborations to continue providing RIVET routines for their measurements.

The parameter points to scan are selected by specifying a resolution for each parameter, which indicates the step size between points on the axis corresponding to that parameter. For each parameter point in the resulting grid, typically 30 000 collider events are generated, each for up to three beam energies (7, 8 and 13 TeV). The event-generation tasks for each point takes approximately one hour, depending on the model under study, and are typically performed in parallel in a high-performance computing (HPC) cluster. In a large grid, where many points are well within or well outside the exclusion contours, this results in hundreds of hours of wasted compute, even for a simple two-dimensional scan.

For a higher resolution of contour line, a larger grid should be generated, which scales with $O(n^k)$, where $k$ is the number of parameters and $n$ is the number of points along each axis. A higher resolution for the grid enables further restriction of the location of the contours, such that a smaller fraction of the total grid must be sampled. Therefore, the sample space should be adapted to the more localised uncertainties.

By iteratively and adaptively sampling the space, the most informative points can be requested from CONTUR–those where we are most uncertain if the point is on one side or the other of an exclusion contour. This uncertainty can be quantified using the *entropy* [24] of each point, as outlined in Section 2.4. This technique is known as active learning [21]. As a result, the total number of sampled points required is significantly reduced, and similar results are achieved. After each iteration, the ORACLE is able to provide an improved prediction of the exclusion level for the whole parameter space and achieves a fast convergence, as discussed in Section 3. An overview of this workflow is given in Figure 1.

## 2.2 Similar tools

Similar applications of machine learning and adaptive sampling can be found in other particle physics re-interpretation tools: GAMBIT [25] uses adaptive sampling to identify viable regions in BSM parameter space from astrophysical and particle-physics inputs; SModelS [26, 27] uses algorithmic techniques to identify new-physics models which are compatible with LHC exclusions on simplified models; the Excursion project [28] uses Bayesian optimisation to attack the problem of efficient scanning of parameter-space. These tools use direct searches as inputs, unlike CONTUR, which uses the bank of LHC model-independent cross-section measurements. The CONTUR ORACLEintroduces a novel method for predicting the exclusion status of new-physics parameter

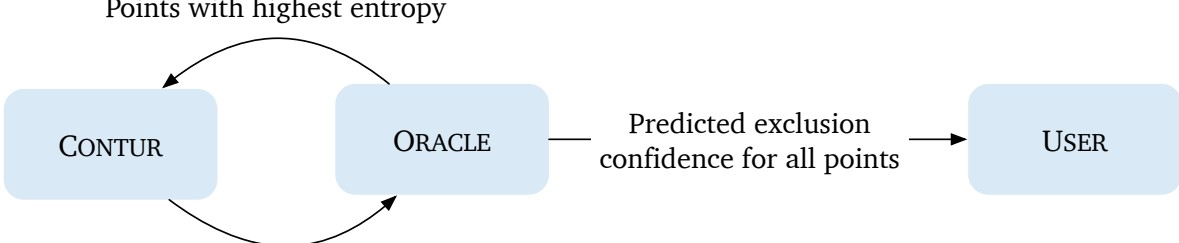

Figure 1: High-level overview of the CONTUR ORACLE workflow. The standard CONTUR programme is used to determine $CL_s$ exclusions for points in the parameter space. This information is provided to the ORACLE module which trains a random forest classifier to make predictions of the $CL_s$ exclusions for arbitrary points. Those points with the highest entropy—a measure of lack of confidence of the classifier in its predictions— are then sampled by CONTUR and the process is repeated until pre-defined stopping conditions are met. The $CL_s$ exclusions for a grid of points in the parameter space are then returned to the user.

points, using random forests.

## 2.3 Training algorithm

An initial low-resolution scan is provided by CONTUR where the points are selected at random from the grid. Each parameter point in the CONTUR scan has an associated CL value, a real number from 0 to 1. Since we are only interested in knowing to which category the parameter point belongs, we convert this continuous variable to a discrete variable $L$, such that $L = 0$ for 0-68% CL, $L = 1$ for 68-95% CL, $L = 2$ for >95%.

In order to verify the accuracy of the predictions, a fraction of the points are selected at random and kept aside in a testing pool. While the training pool is used to train the ORACLE, the testing pool is used to monitor the performance of its predictions.

After training, the machine learning algorithm gives a probability for each exclusion category for each point in the entire grid. The predicted category is simply the one with the highest probability, $\arg\max_{L_i} p(L = L_i)$. In each iteration, the next sample of points to be scanned by CONTUR is selected based on which points would be most informative for the learning process, and which will lead to the greatest reduction in uncertainty. There are several metrics which can be used to quantify this, as discussed in Section 2.4. Other performance metrics are also calculated for the testing and training pools, and determine the stopping conditions which dictate whether the ORACLE should continue sampling or provide a final prediction. During benchmarking, the ORACLE is tested against known CONTUR results, so these metrics are also obtained for the rest of the grid. A summary of this procedure is provided in Figure 2.

The number of points selected in the first and subsequent iterations, the fraction of testing and training points, as well as the stopping condition thresholds, as in Section 2.5, and the number of trees in the random forest, are hyper-parameters which can be adjusted by the CONTUR user. An analysis of the effect of the hyper-parameters in the accuracy and uncertainty of the results is provided in Section 3.4.

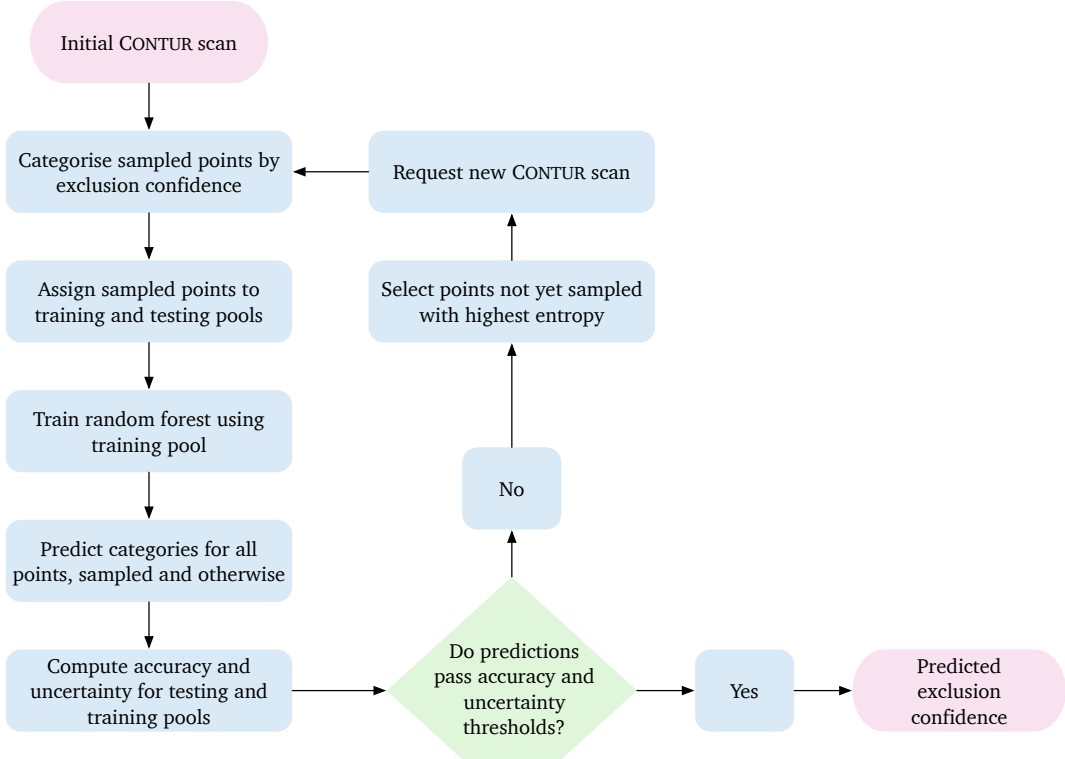

Figure 2: A detailed cartoon of the CONTUR ORACLE workflow. From a grid of parameter points, an initial subset is chosen at random and evaluated using the standard CONTUR programme. The CONTUR results allow the points to be categorised into bins of $CL_s$ exclusion. The points are then split randomly into training and testing pools: the training pool is used to train a random forest classifier to predict the $CL_s$ exclusion bin of a given point. Predictions are then made for all points in the initial grid. Those points which were assigned to the testing pool, for which the true $CL_s$ exclusion bin are known, can be used to test the performance of the classifier. If the performance does not reach a set of predefined conditions, the points with the highest entropy (those for which the classifier is least certain of its predictions) are sent back to the CONTUR programme so that the true $CL_s$ exclusion can be determined, and they can be included in the next training. This process is repeated until the stopping conditions are met, and the predicted $CL_s$ exclusion bins for the whole grid are made available to the user.

## 2.4 Performance metrics

Three metrics are used to measure the performance of the ORACLE and determine when to cease sampling. These are summarised in Table 1.

Different metrics can be used to quantify the uncertainty of a classifier at a given point in the parameter space, such as the smallest margin uncertainty, the largest margin uncertainty, the least confidence uncertainty and the entropy. These metrics can be used to determine which point to sample next by selecting those where the classifier is most uncertain. For the ORACLE, entropy

Table 1: Metrics utilised to determine the performance of the ORACLE

| Metric | Name | Description |
|--------|------|-------------|
| $S$ | Entropy | Uncertainty of the prediction |
| $P$ | Precision | Likelihood of an exclusion prediction being correct |
| $R$ | Recall | Likelihood of an excluded point being detected |

was chosen as a metric for uncertainty:

$$S = -\sum_{l \in \mathcal{L}} p(L = l) \log_3 [p(L = l)], \tag{1}$$

given some probability distribution $p(L)$, and classes $\mathcal{L} = \{0, 1, 2\}$, for a given point in parameter space, as provided by the classifier after each training step. The logarithm in base 3 normalises the range of the entropy to $S \in [0, 1]$ for three prediction classes. The entropy is maximal when the probability distribution is uniform, that is $p(L = l) = \frac{1}{3}$ for all $l$, and minimal when all of the probability mass is in a single category, that is $p(L = l) = 1$ for some $l$ and zero otherwise. To determine when to cease sampling, an arithmetic mean can be computed for the entropy of the points in the testing or training pools.

Measuring the accuracy does not suffice to evaluate the performance of the algorithm. In data sets where only a small fraction of the points are excluded, the algorithm could obtain a high accuracy without detecting the true exclusions, and without providing reliable predictions. In evaluating performance, one should therefore account for how reliable an exclusion prediction is and how sensitive the algorithm is to detecting the underlying exclusions. These two metrics are known as the precision (or positive predictive rate) and the recall (or sensitivity) respectively. For binary classification problems, the precision, P, and recall, R, are defined as:

$$P = \frac{TP}{TP + FP}, \qquad R = \frac{TP}{TP + FN}, \tag{2}$$

where TP is the number of true positives, TN is the number of true negatives, FP is the number of false positives and FN is the number of false negatives.

Since this is a multi-class classification problem, these metrics have to be adapted for more than two classes. We calculate the recall and precision for the 68% and 95% CL thresholds as follows: The recall is the fraction of points which were predicted to have at least 68% CL (95% CL) from the points with a true CL value in 68-95% (95-100%), and the precision is the fraction of points whose true CL was at least 68% CL (95% CL) from the points with a CL predicted to lie in 68-95% (95-100%).

## 2.5 Stopping conditions

After each iteration, once the random forest has been trained, the ORACLE produces predictions for the exclusion status of the points in both the training and testing pools. If the entropy is sufficiently low, and the precision and recall are sufficiently high for both the 95% and 68% CL contours, a final prediction is given. Otherwise, the adaptive sampling continues. If the stopping conditions are never met, the ORACLE gives a final result once it has sampled all the points.

The thresholds for these metrics are user-specified hyper-parameters of the model. The default threshold for entropy is is around $S = 0.2$, for which approximately 95% of the probability mass is

in a certain prediction category. Similarly, the thresholds for the precision and recall are $P = 0.90$ and $R = 0.90$.

## 2.6 Choice of infrastructure

When selecting a machine learning architecture for the classification task, we considered several possibilities. The main criterion for architecture choice was the ability to perform reliably using small data sets from the BSM theory to be probed, and obtaining equivalent results to a full CONTUR scan.

Training a general model which could be applied to different BSM theories was attempted using neural networks, but this proved impractical due to the computational constraints imposed by CONTUR on the number of examples, and the independence of the dynamics and phenomenology of different BSM theories. This motivated the choice to train a classifier for each BSM theory, and the following desiderata. The model should (a) take in continuous numerical features, and output a categorical prediction with an associated probability distribution, and (b) produce interpretable results, such that particle physicists are able to identify the effect of each model parameter on predictions and verify that the predictions are consistent with the BSM theory studied.

In addition, we are only interested in a specific class of functions that describe the exclusion results. For each parameter point, there is often a dominant analysis pool that drives the exclusion. Each analysis pool contains measurements from experiments with similar characteristics, and is only valid for certain intervals of each parameter, as values outside those ranges lead to phenomena not detectable in that class of measurement (see Fig. 3). Therefore, exclusion results are uniform below and above a certain threshold or collection of thresholds, and the boundary between these regions defines the desired hyper-surfaces.

Decision trees are a good choice for classifying the exclusion status of each point, as they split the parameter space precisely according to the boundaries which best fit as hyper-surfaces. However, single decision trees are very susceptible to overfitting, and are not able to provide a metric of uncertainty of a prediction. This motivates the choice of random forests, a collection of decision trees where each tree only has access to a random selection of features and entries from the data set. The final prediction is produced by aggregating the predictions of individual trees, with the probability for each category determined by the fraction of trees predicting that category. Random forests are able to learn with relatively small data sets and model many features, and are easily interpretable by inspecting the individual decision trees and aggregating the most predictive features or conditions. The SCIKITLEARN [29] `RandomForest` class was used in the implementation of the CONTUR ORACLE.

# 3 Benchmarking

To gauge the performance of the ORACLE, a large multidimensional grid for several models was processed by brute force, so that the true $\text{CL}_s$ value—as determined by CONTUR—was known at each point. In each case, three or four of the parameters were varied, while the others were kept constant or defined relative to the varied parameters. The full-grid precision, recall and entropy values could therefore be tracked against the number of points sampled by the ORACLE algorithm.

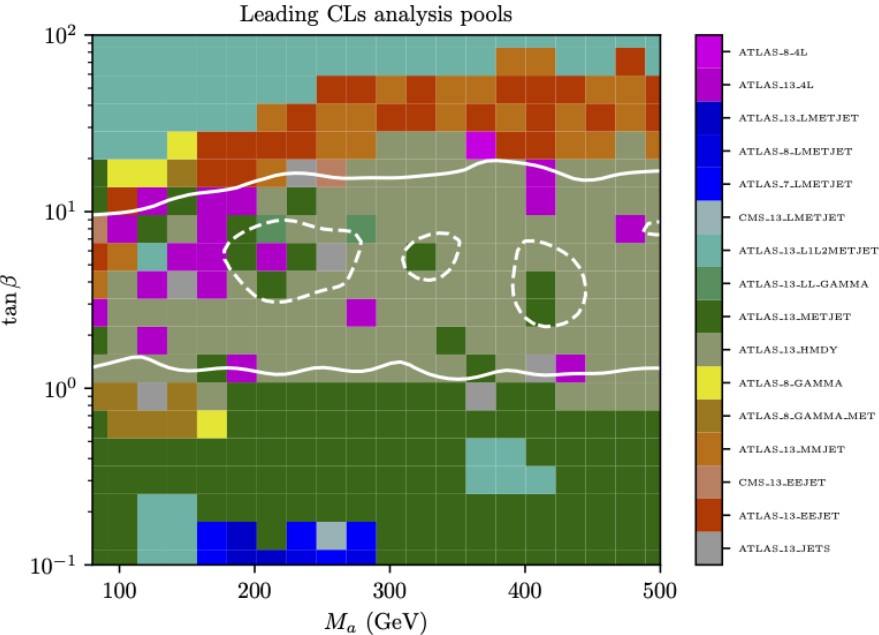

Figure 3: Discontinuities in the dominant analysis pools from the Two Higgs-Doublet Dark Matter Model with Pseudoscalar Mediator. The solid and dashed white lines represent the true 68% and 95% CL contours exctracted from CONTUR while the colours represent the "dominant ananlysis pool": type of analysis (in orthogonal pools of experiment, center-of-mass energy and final state) which gave the best standalone exclusion. The fact that the dominant analysis pool shows discontinuities, linked to the phenomenology of the model under study, which are closely linked to the positions of the exclusion contours, shows that a classifier based on the parameters of a model can be used to extract information such as the exclusion status of a model.

## 3.1 Models used for benchmarking

The performance of the ORACLE was tested on three models which have been studied in past CONTUR publications. Their main features are summarised below. The three models used in the benchmarking process were chosen because they have very different structure, dynamics and phenomenology, and therefore illustrate the ORACLE's versatility.

**Simplified Dark Matter Model with Vector Mediator**    A simplified model for dark matter (DM) [30,31] is used as a benchmark for the LHC collaborations to ease comparison of experimental search results. In the version used for the ORACLE benchmark studies, it features a DM candidate $\chi$ and an extra spin-1 particle $Z'$, which mediates interactions between $\chi$ and SM particles. The model used in this paper has 4 free parameters: the masses of the DM candidate and the mediator ($m_\chi$ and $m_{Z'}$), and the coupling strengths for the $Z'$ with $\chi$ and the SM quarks ($g_\chi$ and $g_q$ respectively), which in this case are assumed to be the same for all quark generations. This model was studied in the first CONTUR publication [14].

**Two-Higgs Doublet Model with Pseudoscalar Mediator**    The two-Higgs doublet model with pseudoscalar mediator (2HDM+a) [32,33] is another benchmark for LHC DM searches. It en-

capsulates the simplest theoretically-consistent extension of the SM which involves DM and a pseudoscalar mediator. In this model, four additional Higgs bosons are introduced (in addition to the known 125 GeV neutral scalar Higgs boson $h$). These are denoted by $H$ (a heavier version of $h$), $H^{\pm}$ (the charged Higgs bosons) and $A$ (pseudoscalar Higgs boson). Furthermore, there is a DM candidate and a mediator pseudoscalar denoted by a. The masses of each of these six new particles are parameters of the model. The coupling between the DM candidate and $a$ is denoted by $g$. Various mixing angles between these states are denoted by $\alpha$, $\beta$ and $\theta$, and by convention the free parameters in the model as expressed as $\tan\beta$, $\sin\theta$ and $\sin(\beta - \alpha)$. Finally three $\lambda$ parameters denote the quartic couplings of the Higgs potential. This model was studied using CONTUR in [34].

**Vector-like Quark Model**   The vector-like quark (VLQ) model introduces four quark partners, $B^{-1/3}$, $T^{2/3}$, $X^{5/3}$ and $Y^{-4/3}$ which couple to the weak sector via vertices linking a VLQ, a SM quark and a $W$, $Z$, or $H$ boson (for $B$ and $T$ only, since $X$ and $Y$ cannot interact with $W$ and $Z$ due to charge conservation). These particles may be arranged in various multiplets depending on how the model is set up. The masses of these four VLQs are parameters of the model. We use the phenomenological framework introduced in [35] where the couplings of the VLQs with SM quarks and bosons are controlled by an overall coupling $\kappa$. Three parameters $\xi$ control the relative couplings to $W$, $Z$ and $H$ (the sum of the three $\xi$ parameters should always be one). Finally, several $\zeta$ parameters control which generation of SM quarks each VLQ can couple to. This model was studied using CONTUR in [36].

## 3.2   Benchmarking approach

For each model, three or four parameters were varied, while the others were fixed or defined as functions of other parameters. The size of the grids for each model was 8 100, 7 260, 3 952 points for the DM, 2HDM+a and VLQ models respectively. For each point in the grid, event generation was performed at 7, 8 and 13 TeV, which tripled the number of event-generation jobs to be submitted to the HPC cluster. For each job, 30 000 simulated signal events were generated. Each job took on average between half an hour, for the 2HDM+a jobs, and nearly two hours, for the VLQ jobs. The total number of wall-time hours required to process these grids was therefore of the order of 17 600 hr, 12 000 hr and 17 200 hr for the DM, 2HDM+a and VLQ models respectively. Creating one-off data sets for benchmarking purposes is possible in some rare cases, but this approach does not scale. The problem only worsens when incorporating even more parameters.

To perform the benchmarking studies, the CONTUR ORACLE was modified to retrieve the $\text{CL}_s$ information for a given point from the aforementioned brute-force grids, instead of sending the job to a batch farm. The other aspects of the workflow stayed the same, and the default hyper-parameter values were used. No stopping conditions were applied, so that the performance of the ORACLE could be studied all the way up to use of the full data set.

## 3.3   Benchmarking results

The results of the study are presented in Figure 4 showing precision, recall and entropy at each iteration. To visualise the size of statistical uncertainties due to the random splitting of test and training samples, after the nominal training the splitting and training were repeated 30 times for each iteration, with the mean and standard deviation shown on the figure.

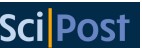

(a) DM

(b) 2HDM+a

(c) VLQ

Figure 4: Performance metrics of the CONTUR ORACLE as a function of fraction of the full-grid points used, for the 95% and 68% CL contours. Given the random nature of the initial sampling, 30 simulations were run; the grey clouds represent the standard deviation. "Full grid" refers to the classifier performance when tested against the "true" exclusion for all the points in the grid (not normally available to the ORACLE unless the full grid was sampled)."Testing" refers to results comparing only to points in the testing pool.

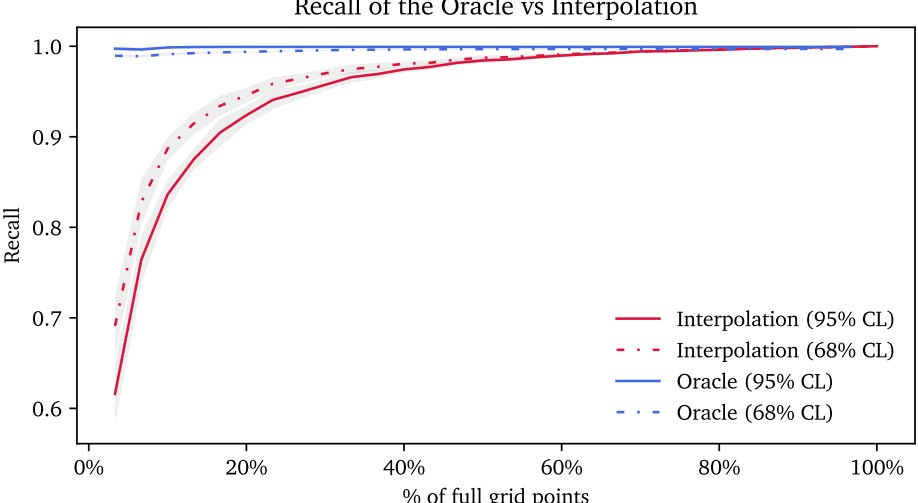

Figure 5: Recall of the CONTUR ORACLE compared to the recall of a simple linear interpolation from a random sample of the same number of points, for the 95% and 68% CL contour of the 2HDM+a model. The ORACLE achieves immediate near-perfect recall, while almost 80% of the parameter points are required for equivalent interpolation results. The grey clouds represent the standard deviation of the 30 simulations.

The behaviour of the ORACLE is remarkably similar for all models, despite the very different phenomenologies which they represent. In all three cases, the precision and recall are above 90% even with the initial sampling of points, both when evaluated on the testing pool and the full grid, and for both 68% and 95% CL exclusion predictions. The full-grid precision and recall increase at each iteration, while the testing pool precision and recall suffer small dips before steadily rising. This is explained by the fact that the sampling algorithm deliberately selects the points where the predictions have the highest entropy: in other words, the ones where it is hardest to correctly predict the result. There is therefore a lag time before the testing pool performance improves again.

The entropy for all three models also follows a similar trend. The full-grid entropy steadily decreases (this makes sense since this is the figure of merit for the adaptive sampling), while the entropy of the testing pool peaks in the first few iterations before dropping steadily. This can be explained by the fact that with little data, the ORACLE initially can be quite confident in its incorrect predictions, before revising them as more points are added.

In all three models, after sampling only one fifth of the points in the grid, the ORACLE was able to achieve a full-grid precision and recall of at least 99%. The performance of the ORACLE against a simple interpolation algorithm, in terms of recall, is shown for the 2HDM+a model in Figure 5, showing the gains from the machine-learning assisted adaptive sampling compared to simply interpolating between results.

An illustration of the convergence of the ORACLE for the 2HDM+a model is shown in Figure 6, in a slice of the parameter grid described above. The figure shows how the ORACLE's predictions converge towards the standard CONTUR results, and matches them perfectly after only 5 iterations, with 1 500 of the total 8 100 points sampled.

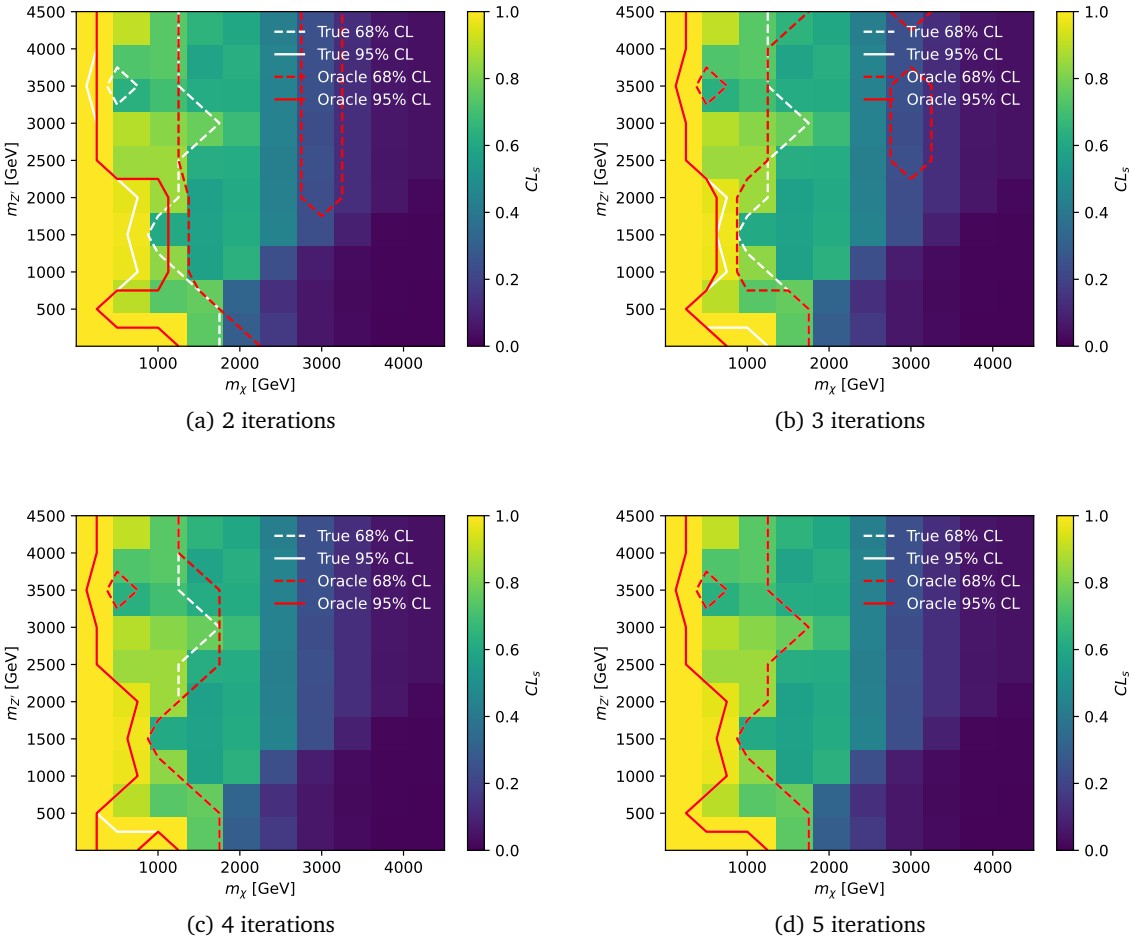

Figure 6: Comparison of the 95% (solid lines) and 68% (dashed lines) exclusion contours from the brute-force CONTUR scan (in white) and from the ORACLE predictions (in red) for 2-5 iterations, for a slice of the four-dimensional grid of parameter points for the simplified DM model where $g_\chi = 0.65$ and $g_q = 0.15$. In this slice, after five iterations (or less than 20% of the total grid), the ORACLE reproduces the standard CONTUR results perfectly.

## 3.4 Effect of hyper-parameters

Using the simplified DM model, the effect of the choice of hyper-parameters on the ORACLE's performance was investigated. There are three major hyper-parameters available to modify: the number of trees in the random forest, the number of points per iteration (effectively, the batch size for the adaptive sampling) and the fraction of points used for the test sample. The default values for these hyper-parameters in the ORACLE are 300 trees, 300 points per iteration and $\frac{1}{4}$ of the pool used for testing. The same methodology as in Section 3 is used, but with a range of different values of these hyper-parameters. It was found that the performance of the ORACLE does not depend strongly on the exact choice of hyper-parameters.

Batch sizes between 150 points and 900 points were tested, representing iterations of between

2% and 11% of the overall grid size. The full-grid precision and recall converge to high values faster for low batch sizes, as expected, since there are more opportunities to select difficult points before reaching high fractions of the data set. The entropy of the full grid also drops faster for smaller batch sizes. In reality, there is a trade-off to be made between small batch sizes and the advantage of being able to run more points in parallel in a HPC farm. Indeed, although low batch sizes allow the ORACLE to converge fastest in terms of fraction of the overall data set, it may be faster in terms of user wall time to run large batch sizes, where more points can be sampled in parallel. The developers would recommend the default setting of a batch size be set to approximately half the number of jobs which can be run in parallel by the user's HPC farm, divided by three if each point has to be sampled for three beam energies. This can be adjusted depending on the user's requirements: lower batch sizes lead to lower compute requirements, but more wall time before convergence, and vice versa.

The number of trees for the random forest does not have a strong effect on the convergence of the ORACLE as long as the number is much higher than the number of parameters in the model. All choices for this hyper-parameter gave comparable results, except for the extremal case of 1 tree, where the performance was obviously degraded due to insufficient flexibility of the model. The ORACLE training time increases for larger numbers of trees (up to a few seconds), but this is negligible in the face of time taken to sample grid points. The developers therefore recommend the default setting of a few hundred trees.

Finally, the testing–training fraction has a small effect on the performance of the model: typically, the smaller the test fraction, the better the performance, since the model can be trained with more of the available points. However, the trade-off is that small testing fractions lead to large statistical uncertainties in the testing pool accuracy, which is one of the stopping conditions for the ORACLE. The risk is therefore that the ORACLE workflow terminates prematurely if a too-low testing fraction is chosen. Testing fractions between 12.5% and 25% gave similar results. The default of 25% is deemed by the developers as an appropriate compromise between statistical fluctuations in the stopping conditions and final performance.

## 3.5 Prediction uncertainty

There are two ways that we have used to measure the unceratinty of the ORACLE predictions. First, a fraction of the points received from each iteration are kept aside for validation purposes, and the uncertainty of the predictions can be measured by comparing the ORACLE's predictions for all the points in the testing pool and the CONTUR results through the precision and recall testing metrics, which are shown in Figure 4. In addition, the uncertainty of a certain prediction can be measured by inspecting how concentrated the probability mass is on the predicted class. The entropy of the distribution is one such measure of this, also present in Figure 4.

The reliability of the ORACLE's predictions can be customised by configuring the precision and recall stopping conditions, and by modifying the threshold for declaring a point to belong to a certain class, thus leveraging the trade-off between precision and recall. In this paper, we have used the convention of declaring the prediction to be the class with the highest probability, but this can easily be modified to require a higher probability threshold to declare an exclusion, and declare a non-exclusion otherwise, or vice-versa. A high exclusion threshold will lead to a higher precision of the predicted exclusions, but will also miss many exclusions and miscategorise them as non-exclusions. Conversely, a low exclusion threshold will achieve the opposite effect.

Users can leverage this trade-off between precision and recall by modifying the threshold according to their needs. Therefore, the ORACLE results are intended to be used as an approximation

of the CONTUR results that can optimise for the metrics of interest to the user without requiring nearly as much computational power, rather than strict exclusion boundaries.

Two other methods have been used to measure uncertainty in the benchmarking section: first, running several independent ORACLE instances in parallel, and then obtaining the average and standard deviation of the results, as can be seen in Figure 4. Second, the results of the Oracle were compared with the full-grid CONTUR results. These methods are only currently useful for benchmarking and not for analysing a novel BSM model, but they can easily be modified for this purpose. Boostrapping or cross-validation techniques can be used to perform the train/test split multiple times, and separate classifiers can be trained for each split. Their predictions can be averaged for selecting the next set of points to sample. This was manually done for the novel model in Section 4, but is not currently implemented in the software. On the other hand, a fraction of the sampled points in each iteration can be a random sample of the full grid, which can be used to measure the uncertainty of the ORACLE predictions, since the uncertainty from the training points is a considerable over-estimate as the algorithm intentionally selects points that are difficult to classify.

Including these methods in the ORACLE is technically possible, but is left as future work.

# 4 Application of the ORACLE to a new model

The CONTUR ORACLE was further used to study a model at a previously impossible dimensionality and granularity, to provide a real-world example of its versatility. For this test, a more complex version of the simplified DM model described Section 3.1 was used, in which the couplings of the vector mediator to quarks can be set separately for each generation. The coupling of the DM candidate to the vector mediator was fixed to 1, but the five other parameters were varied independently. The couplings of the vector mediator to first-, second- and third-generation quarks ($g_{q1}$, $g_{q2}$ and $g_{q3}$) were all varied independently between 0.09 and 0.89 with a resolution of 0.1, leading to 10 points on each axis. The mass of the DM candidate was varied between 10 GeV and 4 760 GeV with a resolution of 250 GeV, leading to 20 points on that axis. Finally, the mass of the vector mediator was varied between 10 and 9510 GeV with a resolution of 500 GeV, leading to 20 points on that axis. The total number of points on the grid was therefore $10 \times 10 \times 10 \times 20 \times 20 = 400\,000$, which would cost nearly half a million hours of compute time using CONTUR. This type of scan would previously have been impossible. The default ORACLE parameters were used, and the stopping conditions were set at 90% for precision and recall, and 0.2 for entropy. The scan was limited to 13 TeV exclusions.

The ORACLE run reached the stopping conditions after sampling just over 25 000 points—6.3% of the total grid size—of which approximately 8 000 were used for testing. At that stage, the precision was found to be 90%, the recall 92% and the entropy 0.11. Figure 7 shows a sample of the results from this scan, showing exclusion contours and entropy at each point. To obtain a measure of uncertainty in the results, we later performed bootstrapping, i.e. repeated train/test splits of the sampled points in each iteration, and trained a separate classifier for each split. We let the ORACLE run until it sampled 12% of the full grid, and the precision was found to be $(94.7 \pm 0.5)$% for 68% CL, $(92.8 \pm 0.4)$% for 95% CL, the recall $(94.3 \pm 0.5)$% for 68% CL and $(94.9 \pm 0.5)$% for 95% CL, and the entropy $0.057 \pm 0.002$, where the quoted uncertainties refer to the standard deviation of the sampled statistic.

The exclusion depends chiefly on the coupling of the vector mediator to first-generation quarks:

the higher the coupling, the higher the excluded mediator mass $m_{Z'}$. The dependence on the DM mass is typically quite flat. For example, for $g_{q1} = g_{q2} = g_{q3} = 0.49$, vector mediator masses up to about 3 500 GeV are excluded at 95% CL, and up to nearly 5 000 GeV at 68% CL. If the value

Figure 7: Examples of ORACLE exclusion contours extracted from 13 TeV LHC measurements, in slices of the parameter space of a simplified DM model with five free parameters. The testing pool recall, precision and entropy were 0.9, 0.92 and 0.11 respectively, after sampling only 6.3% of the total grid size. The results are presented as a function of the dark matter mass ($m_\chi$) versus vector mediator mass ($m_{Z'}$), for fixed values of the couplings of the vector mediator to the three generations of SM quarks, $g_{q1}$, $g_{q2}$ and $g_{q3}$. The colour scale shows the entropy for the prediction at each point, which is a measure of the uncertainty of the prediction: higher values indicate regions where the prediction is more uncertain. The red dashed and solid lines represent the 68% and 95% exclusion contours predicted by the ORACLE.

of $g_{q1}$ is reduced to 0.09, the exclusion contours retreat to about 2 000 GeV; while if the value of $g_{q1}$ is increased to 0.99, the exclusion contours increase to about 5 500 GeV. This can be explained by the fact that CONTUR uses analyses of data collected from proton–proton collisions, where the first-generation quarks dominate the parton distribution functions. Therefore, the vector mediator production cross-section will depend strongly on its coupling to the valence quarks, which is $g_{q1}$, especially for high mediator masses.

When $g_{q1}$ is maintained at 0.49 but the other two couplings are varied between 0.09 and 0.89, either together or separately, the contours shift only by about 500 GeV at most. Similar flat dependence on the $g_{q2}$ and $g_{q3}$ couplings is observed when $g_{q1}$ is set to high values. However, when $g_{q1}$ is set to low values, such as 0.09, the exclusion gains a dependence on $g_{q3}$, while remaining largely independent of $g_{q2}$. Indeed, for $g_{q1} = 0.09$ and $g_{q2} = 0.49$, the exclusions in vector mediator mass vary from about 2 500 GeV for high values of $g_{q3}$ to 1 000 GeV or less for low values of $g_{q3}$. This dependence occurs because in the low-$g_{q1}$ but higher-$g_{q3}$ regime, measurements of processes involving top quarks contribute to the exclusion.

The entropy at each point is also shown in Figure 7. As expected, the regions with high entropy are concentrated chiefly in the regions near the exclusion contours. This indicates that outside of these regions, the ORACLE is able to quickly determine the exclusion status, using few points for most of the parameter space away from the exclusion hyper-surfaces.

This exercise further demonstrates the power of the CONTUR ORACLE to provide insights on the variation of the LHC exclusion of a model with its parameters, using reduced computing resources: in this case, this corresponds to an improvement by a factor of 16 compared to the case where all points are sampled.

## 5 Conclusion

This paper introduces a method to focus computing resources on the most interesting regions of parameter space when evaluating the exclusion status of new physics models using re-interpretation tools such as CONTUR. Our implementation is publicly available as part of CONTUR v2.2.0.

By combining adaptive sampling with random forests, the method achieved precision and recall scores of 90-95% for a variety of models with different phenomenology and underlying dynamics, using no more than 5-10% of the data points and computational resources required for a full scan using state of the art methods such as CONTUR. To demonstrate the method, a study of a model in a dimensionality and granularity which would otherwise have been impossible was conducted, and the key features of the exclusion in that space are successfully summarised using only a sixteenth of the computational resources that would otherwise have been required.

While the performance of the method is already impressive, this is only a first step. The exclusions obtained from each measurement pool could be provided to the algorithm, instead of the total exclusion, which would produce even more accurate predictions. This would also shed light on which types of measurement yield the best predicted exclusion for any given point. We believe that harnessing the power of adaptive learning can play a key role in determining the models and regions which lead to future discoveries at the LHC, and that this method can be used to design searches and measurements to study those regions. This method relies on the publication of RIVET routines for new LHC measurements, and we encourage the collaborations to continue making ever more of their measurements available in this format.

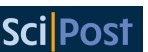

# Acknowledgements

**Funding information** JR received funding from the Barrie Foundation Fellowship for postgraduate studies at the University of Cambridge. This work has received funding from the European Union's Horizon 2020 research and innovation programme as part of the Marie Skłodowska-Curie Innovative Training Network MCnetITN3 (grant agreement no. 722104).

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
