# Peer review of "Picking the low-hanging fruit: testing new physics at scale with active learning"

_SciPost Physics Core, doi:SciPost Phys. 13, 002 (2022)_

## Round 1 · Referee Report · Anonymous (Referee 1) · 2022-3-8

Strengths

1- To my knowledge, this paper is tackling an issue which is definitely overlooked in our field. It is something that deserves more attention.

2- The paper is very well written and easy to read.

3- Adding a novel model for testing was a good idea

Weaknesses

No major weaknesses.

1- The paper lacks (in some parts more than in others) a comprehensive set of citations. For authors expert in the field, as these are, it's not much work since you do not have to find the sources, you have them and you usually fix this problem very easily by spending a boring 2 hours filling in all the places where a citation is missing.

2- The paper was clearly proof-read but it requires another two or three runs. There are still grammar errors and typos and some figures are missing a proper caption

Report

This is a very interesting paper on an interesting and relevant subject written by someone that knows the English language. Well done. Being a HEP physicist myself I find the paper well written and easy to understand. It requires some editorial work detailed below but I would say nothing major.

The main part that I would add is a proper and detailed study of the uncertainties related to the contours that you are proposing e.g. in figure 9. E.g. in GAMBIT it is well knows that they use a Poissonian likelihood marginalised over a rescaling parameter to account for systematic uncertainties. What do you do?

Another point that I would like to raise is your dependence on the RIVET project. If I understand correctly, you need an analysis to be on RIVET in order to use you tool on it. This is a major shift in the tone you are using in Section 1. While I understand the importance of getting the reader to understand that you code "matters" I think the relatively small ratio of ATLAS, CMS, and LHCb analyses available on RIVET should be acknowledged as a bottleneck.

For the rest, well done and thanks!

Requested changes

The changes that are outlined in the weakness and report sections:

Majors: 1- Can you try to add a paragraph about uncertainty handlings or the lack of it? E.g. like GAMBIT use of a Poissonian likelihood marginalised over a rescaling parameter to account for systematic uncertainties. If you do not have this I am not saying you should modify your code but acknowledge the issue.

Minors: 1- Increase citations (very obvious in Sec 1)

2- Tone done a bit Sec 1 or explain the problem with RIVET availability

3- Proofread another two or three times, there are typos and grammatical errors, you are clearly native speakers so this should not take long

4- I would remove Figure 1 which is not adding any information (but do as you like)

5- You need better Figure captioning across the paper

6- Avoid footnotes

7- Equations should be numbered

8- In some figure’s axes are not labelled

9- Section 3.4 is a bit too qualitative to my taste, here a good quantitative paragraph about systematic uncertainties could be a welcomed addition

  • validity: high
  • significance: high
  • originality: high
  • clarity: high
  • formatting: good
  • grammar: excellent

Author:  Juan Rocamonde  on 2022-04-13  [id 2380]

(in reply to Report 1 on 2022-03-08)

We thank the referee for his supportive review and helpful comments, which we have taken on board. We have made the following changes to the manuscript:

  • Added a new Sec 3.5 which has a discussion about the uncertainties in this method and how they can be assessed. Although the code we have written does not handle these automatically at present, they are acknowledged and discussed, and provided with recipes for how they can be evaluated.
  • We added citations in relevant places, especially in Sec 1.
  • We have added in Sec 1, Sec 2 and the conclusion sentences which make the relationship between CONTUR and RIVET clear, and use the opportunity to call for ever more LHC measurements to be published as RIVET routines.
  • As suggested, we’ve done a few further iterations as well as passed the text through spell-check and grammar-check programmes, making a few changes as a result.
  • Equations have been numbered and integrated the footnotes in the main text, and expanded captions for many figures.

We think these modifications have improved the manuscript and we are thankful for the comments which have led to these improvements. We hope the referee agrees!

---

## Round 1 · Referee Report · Anonymous (Referee 2) · 2022-3-9

Strengths

1) Well-written 2) Relevant 3) Timely 4) Reasonably comprehensive for this kind of thing 5) Insightful 6) Substantive progress in the field

Report

This is an excellent paper that meets the journal's criteria. I recommend it be published. There is very little to criticize because it is lucid and complete.

Requested changes

I only spotted a few typos:

Page 3, second-to-last paragraph:

-- "and the results distribution are compared to the observed results for LHC analyses"

Page 4, first full paragraph"

-- "locstion"

Page 8, last paragraph:

-- "resptectively"

Page 9, second paragraph:

-- "prameters"

Page 11, last partial paragraph:

-- "this make sense"

  • validity: high
  • significance: high
  • originality: high
  • clarity: high
  • formatting: excellent
  • grammar: excellent

Author:  Juan Rocamonde  on 2022-04-13  [id 2379]

(in reply to Report 2 on 2022-03-09)

We thank the referee for their positive assessment, and for directing us to the typos, which we have now fixed, and apologise for missing.

We have also re-run a spellcheck, and caught a few other minor typos and repetitions.

---

## Editorial Decision

published